Identification and classification of Aquilaria (Thymelaeaceae): inferences from a phylogenetic study based on matK sequences

Xie Zhaoqi 1
Fan Siqing 1
Xu Junyu 2
Xiao Haijing 1
Yang Jiaxin 1
Guo Min 1
Cheng Chunsong chengcs@lsbg.cn 1
1 Jiangxi Key Laboratory for Sustainable Utilization of Chinese Materia Medica Resources, Lushan Botanical Garden, Chinese Academy of Sciences , Jiujiang , China
2 School of Life Sciences, Nanchang University , Nanchang , China
Manjarrez Javier
Electronic publication date: 2025 Jul 23
Publication date: 2025
Volume: 13
Electronic Location ID: e19752
Received 2024 Oct 27; Accepted 2025 Jun 24
Copyright: ©2025 Xie et al.
Copyright year: 2025
Copyright holder: Xie et al.
License: This is an open access article distributed under the terms of the Creative Commons Attribution License, which permits unrestricted use, distribution, reproduction and adaptation in any medium and for any purpose provided that it is properly attributed. For attribution, the original author(s), title, publication source (PeerJ) and either DOI or URL of the article must be cited.
License URL: https://creativecommons.org/licenses/by/4.0/

Keywords: Aquilaria, matK, Molecular markers, Agarwood, Polymorphic loci, Molecular clock

Funding: Jiangxi Province Double Thousand Talent-Leader of Natural Science Project jxsq2023101038 Jiangxi Province Urgently Overseas Talent Project 2022BCJ25027 Key Research and Development Special Project of Jiangxi Province S2023ZPYFB0294 & 20223BBH80007 Science and Technology Innovation Team Project in Key Areas of Jiujiang City Base and Talent Plan S2022TDJS029 Jiangxi Provincial Natural Science Foundation General Project 20242BAB25341 Special Project for Lushan Plants 2023ZWZX07 This work was funded by Jiangxi Province Double Thousand Talent-Leader of Natural Science Project (jxsq2023101038), Jiangxi Province Urgently Overseas Talent Project (2022BCJ25027), and the Key Research and Development Special Project of Jiangxi Province (S2023ZPYFB0294 & 20223BBH80007). This work was also funded by the Science and Technology Innovation Team Project in Key Areas of Jiujiang City Base and Talent Plan (S2022TDJS029), the Jiangxi Provincial Natural Science Foundation General Project (20242BAB25341) and the Special Project for Lushan Plants (2023ZWZX07). The funders had no role in study design, data collection and analysis, decision to publish, or preparation of the manuscript.

==============================
In the realm of Aquilaria classification and grading, a persistent market uncertainty persists, questioning whether the basis should be geographical distribution or biological origin. In this study, the effectiveness of matK molecular markers, particularly through eight stable polymorphic loci (e.g., +249C for Chinese origin, +435G for Aquilaria sinensis), emerges as a decisive tool for differentiating Aquilaria species. The integration of matK and trnL-trnF not only validates this efficacy but also streamlines the systematic categorization of 34 agarwood products into four biogeographic pedigrees: Chinese (C1: A. sinensis; C2: A. malaccensis), Indonesian (A. cumingiana), and Indochinese (A. rugosa). Molecular clock analyses trace the genus’s divergence to 6.78 million years ago (Ma) (A. hirta), with recent speciation of commercially pivotal species (A. sinensis:  0.9 Ma; A. malaccensis:  1.0 Ma). Notably, the redefined placement of Gyrinops walla (5.75 Ma) within Aquilaria challenges prior taxonomic assumptions, suggesting revised genus boundaries. The Median-Joining network further visualized these haplotypes, showing key evolutionary transitions, particularly from A. crassna to A. rugosa and A. malaccensis. These findings provide robust tools for species differentiation, insights into evolutionary history, and practical guidance for conservation and trade applications within the field of botany.

Introduction

Agarwood, also known as gaharu wood, stands as a renowned trade name for a highly prized resinous substance obtained from infected trees, primarily belonging to the genera Aquilaria (A.) and Gyrinops (G.) (Ma et al., 2021; Paoli et al., 2001). While agarwood is also produced by other genera in the Thymelaeaceae family, such as Aetoxylon, Claoxylon, Encleia, Gonystylus, and Wikstroemia, the quality of agarwood from these species is generally considered inferior and less commercially viable due to the presence of fewer or less potent aromatic compounds, particularly sesquiterpenes and lignans, which are key contributors to the characteristic fragrance and market value of agarwood (Li et al., 2023; Yin et al., 2016). Agarwood has diverse applications, ranging from traditional medicine in China and India to its use in the production of perfumes and incense in Japan, the Middle East, and Arabic countries. In many Asian regions, it is also used as a preservative for various goods and accessories (Peeraphong, 2021; Shu-Yuan, 1995). The rising global demand for agarwood has led to increased exploitation, threatening the natural populations of agarwood-producing species (Hashim et al., 2016; Herath & Jinendra, 2023; Tan et al., 2019). The current number of agarwood tree species is far from meeting market demand. Therefore, it is particularly important to protect agarwood tree species, such as developing molecular tools for the accurate identification and conservation of agarwood-producing species, promoting artificial planting, and implementing sustainable management.

The accurate identification of agarwood-producing species is critically hindered by morphological ambiguities in traded products, necessitating molecular approaches to resolve taxonomic and conservation challenges. After nearly a decade of extensive sorting and exploration, a total of 14 species belonging to the genera Aquilaria and Gyrinops have been identified as producers of agarwood (Eurlings & Gravendeel, 2005; Lee et al., 2022). These species predominantly thrive in tropical rainforests stretching from India to Southeast Asia. The majority of agarwood products in the market are derived from eight species: Aquilaria sinensis, A. malaccensis, A. crassna, A. hirta, A. subintegra, A. filaria, A. microcarpa, and G. walla. To better control circulation and harvesting, agarwood products have been classified into three regional groups: the Dongguan group (South China, including Guangdong, Hainan, and Fujian provinces), the Huiangroup group (Indochina Peninsula and Southeast India, including Vietnam, Cambodia, and Laos), and the Xingzhou group (Java, Sumatra, Kalimantan, and Malaysia) (Lee & Mohamed, 2016; Tan et al., 2019). Among the Southeast Asian countries, the most valuable and highest quality agarwood was historically derived from the unique species A. malaccensis. Consequently, the focus of conservation efforts primarily centered on protecting the A. malaccensis species. Initially, Indonesia exported all agarwood under the designation A. malaccensis, despite the presence of agarwood products in the market derived from other Aquilaria species (Lloren, 2023; Mustaqim, 2020). Due to the lack of fruits and flowers, it is difficult to identify the species level of agarwood wood products on the market, which directly increases the difficulty of chemical analysis. This underscores the urgency for molecular identification tools to disentangle taxonomic confusion and enforce protections for threatened Aquilaria populations.

Traditionally, agarwood species identification methods have relied on morphological traits and physicochemical properties. However, these approaches have limitations, particularly in distinguishing between closely related or morphologically similar species, and are often influenced by environmental factors. Without fruit or flowers, it is difficult to identify agarwood products on the market at the species level, which complicates chemical analysis. This has led to market issues such as counterfeit products, substandard goods, and the misrepresentation of product quality, severely disrupting market order and eroding consumer trust. With the advancement of molecular biology techniques, DNA barcoding has emerged as a powerful tool for plant species identification. A DNA barcode refers to a standardized DNA fragment within the genome. Due to its characteristics of minimal intraspecific variation, substantial interspecific variation, and highly conserved sequences at both ends of the variable region, it has gradually been adopted for rapid species identification and authentication. In 2009, the Consortium for the Barcode of Life (CBOL) recommended the combination of the chloroplast genes matK and rbcL as the DNA barcode for land plants, and proposed the chloroplast gene fragment psbA-trnH and the ribosomal DNA internal transcribed spacer (rDNA-ITS) sequence as supplementary barcodes (Chen et al., 2010; Group1 et al., 2009). Global efforts have led to the establishment of a genetic sequence database for identification purposes, most notably the Barcode of Life Data Systems (BOLD), with rbcL and matK being the primary barcodes for plants. Studies by Liu et al. (2019) on bryophytes identified several potential DNA barcodes, including rbcL, rpoC1, trnH-psbA, rps4, and trnL-trnF. The China Plant Barcode Research Group has made significant strides in seed plant DNA barcoding. Their research focused on 75 families, 141 genera, and 1,757 species, analyzing approximately 6,286 samples. The study found that the rbcL, matK, and trnH-psbA plastid DNA barcodes had high universality, while the ribosomal DNA ITS barcode exhibited the highest species resolution. ITS1 and ITS2 also showed promising results for species identification, particularly in angiosperms (China Plant BOLG et al., 2011).

DNA barcoding has already been widely applied to distinguish between commonly confused Aquilaria species, such as A. sinensis, A. malaccensis, and A. crassna (Farah et al., 2018; Lin et al., 2023; Lee et al., 2022). Recent work using multi-locus DNA barcoding (e.g., ITS, matK, rbcL) and complete plastid genomes has significantly advanced the understanding of Aquilaria and Gyrinops relationships. For instance, Lee et al. (2022) resolved the paraphyletic relationship between Aquilaria and Gyrinops using complete plastome data, revealing that Aquilaria nests within Gyrinops. Similarly, Farah et al. (2018) estimated divergence times for Aquilarieae speciation during the Miocene using chloroplast and nuclear DNA regions. In addition, the current DNA barcoding technology overlooks the analysis of plant biogenetic backgrounds within specific geographical regions or populations, leading to concerns about the reliability and validity of the conclusions drawn from such studies (Jiao et al., 2014; Liu et al., 2013; Taberlet et al., 1991; Tanaka & Ito, 2020). By focusing on the matK sequence, which demonstrated exceptional discriminatory power for species identification, this study provided a robust molecular foundation for distinguishing between species based on their genetic diversity. The study’s emphasis on multiple molecular markers (ITS2, rbcL, trnL-trnF) and their comparative effectiveness for species differentiation. By identifying specific polymorphic loci that reliably differentiate species based on their geographic origins, such as the unique thymine base pattern at locus +684 in A. sinensis or the cytosine at locus +249 in A. sinensis and Chinese-origin species, this study provided a practical tool for distinguishing species within this genus. Moreover, the molecular clock analysis revealed a clear timeline of divergence among species, pinpointing key evolutionary events and lineages that align with regional distribution patterns. Four distinct genetic lineages were identified, which correspond to A. sinensis (C1-2), A. malaccensis (B), and A. cumingiana (A), indicating that these species possess multiple independent origins. Finally, by addressing gaps in existing research—such as the limitations of certain markers like ITS2—and introducing novel approaches (the utilization of matK markers or the integration of matK markers with the trnL-trnF region) to evaluate marker performance across the entire species group, this study not only advances taxonomic understanding but also provides actionable insights for conservation and sustainable management of agarwood resources.

Materials and Methods

Portions of this text were previously published as part of a preprint (https://www.researchsquare.com/article/rs-4120659/v1).

Plant materials

The agarwood samples (wood) used in this study were purchased from various commercial sources worldwide (Fig. S1 and Table S1) and were preserved in a silica gel desiccator. To ensure the traceability and verification of the plant materials, voucher samples have been deposited in the public herbarium of Macau University of Science and Technology. A detailed list of the plant materials used, including their ID, full name, and source, is provided in Table S1. The leaves of Aquilaria used for PCR identification were collected from the medicinal herb gardens of Guangzhou University of Chinese Medicine, specifically from Shizhen Mountain and Yaowang Mountain. The parent plant is Aquilaria sinensis (Lour.) Gilg, belonging to the family Thymelaeaceae. The samples were sterilized with 70% ethanol, rinsed with sterile water, air-dried, and stored at −20 °C. Species nomenclature and taxonomic validation were conducted using the online database The Plant List (Version 1.1, http://www.theplantlist.org/), accessed directly via its web interface. For each Aquilaria species, we cross-referenced scientific names, synonyms, and accepted classifications to resolve taxonomic discrepancies. Distribution data were compiled from peer-reviewed literature (Farah et al., 2018; Lee & Mohamed, 2016; Ng, Chang & Kadir, 1997), commercial trade reports, and verified herbarium records. To ensure consistency, species names were standardized against The Plant List entries. Moreover, relevant geospatial data such as latitude, longitude, and altitude for each specimen’s origin were extracted using AMAP (AutoNavi, Version v15.13.0.2021) and Google Earth (Version v1.1.2). These coordinates were then projected onto a provincial boundary map of China using ArcGIS (Version 10.7) software, enabling us to observe and analyze the distribution patterns of different groups. The mapping reconstruction was performed utilizing the ArcMap tool and the Lambert projection method, and the results were saved in TIF/PNG format with a resolution of 300 dpi.

Sample processing and DNA extraction

Agarwood samples were first disinfected with 75% ethanol and then cut into small strips or blocks on clean, new paper. The samples were transferred to 2.0 mL Eppendorf tubes and ground into powder using a Tissuelyser II (Qiagen, Hilden, Germany) in the presence of liquid nitrogen. For samples with high resin content, a Red solution (3M sodium hydroxide with phenolphthalein) was used to soak the tissue. Neutralization was achieved by adding 3M hydrochloric acid once the solution turned brown. In cases of higher resin content, the sample was further ground in the presence of 3M hydrochloric acid using an agate mortar and then transferred to a 2.0 mL Eppendorf tube. For DNA extraction, approximately 100 mg of dried leaf tissue from A. sinensis was selected. The tissue was ground into powder under liquid nitrogen and transferred to a 2.0 mL centrifuge tube. To the powdered tissue, 1,000 µL of DNA wash buffer (100 mmol/L Tris-HCl, pH 8.0, 0.14 mol/L NaCl, 20 mmol/L EDTA) containing 1% β-mercaptoethanol was added. The sample was vortexed for 1 min, centrifuged at 12,000 rpm for 10 min at 4 °C, and the supernatant discarded. Subsequently, 700 µL of pre-warmed 2% CTAB extraction buffer (100 mmol/L Tris-HCl, pH 8.0, 1.4 mol/L NaCl, 20 mmol/L EDTA, 3% CTAB) at 65 °C was added. The sample was vortexed for 1 min and incubated in a 65 °C water bath for 30 min. The mixture was treated with 700 µL of phenol:chloroform:isoamyl alcohol (25:24:1, v/v) and vortexed for 30 s, followed by centrifugation at 12,000 rpm for 10 min at room temperature. The upper aqueous phase was carefully transferred to a new tube, to which an equal volume of chloroform:isoamyl alcohol (24:1, v/v) was added, and the procedure repeated. Finally, the aqueous phase was mixed with an equal volume of pre-chilled isopropanol, gently mixed, and the precipitate collected. The pellet was washed twice with 70% ethanol and once with absolute ethanol, then air-dried at room temperature. The DNA was resuspended in 50 µL TE buffer and stored at −20 °C. Additionally, leaf samples from A. sinensis were processed following the same procedure, with DNA extraction carried out using the DNeasy® Plant Mini Kit (Qiagen) according to the manufacturer’s instructions.

Molecular marker sequence acquisition and analysis

Molecular marker sequence data used in this study were retrieved from the National Center for Biotechnology Information (NCBI, https://www.ncbi.nlm.nih.gov/). The downloaded sequences and their corresponding accession numbers are provided in Tables S2–S5. Sequences of the matK, ITS (ITS2), rbcL, and trnL-trnF markers for each species were obtained from NCBI. These sequences were processed using BioEdit (Version 7.01, available at https://bioedit.software.informer.com/) for alignment and analysis. After trimming low-quality regions with SnapGene® Viewer (Version 2.6.2), raw sequences were aligned in BioEdit using the ClustalW algorithm with default parameters (gap opening penalty = 10, gap extension penalty = 0.1). The software facilitated the identification of conserved and polymorphic loci, including manual corrections for ambiguous bases. Aligned sequences were then exported to MEGA 7.0 for phylogenetic reconstruction (Kumar, Stecher & Tamura, 2016). Sequence alignment was conducted using the Kimura 2-parameter (K2P) model to compute genetic distances, followed by the construction of a Neighbor-Joining (NJ) tree. Gaps were treated as missing data, and bootstrap resampling (1,000 replicates) was applied to assess the confidence of the tree branches. Phylogenetic trees were constructed using the Neighbor-Joining method to evaluate genetic distances between species and markers, with statistical comparisons of inter- and intraspecific distances conducted via t-tests and Kruskal–Wallis tests. In addition, RStudio (Version 2024.09.1) (RStudio Team, 2014), and SPSS (Version 20.0.; SPSS Inc., Chicago, IL, USA) were utilized for the implementation of basic mathematical statistics. t-tests: to compare the interspecific and intraspecific genetic distances, we performed t-tests to determine if there were statistically significant differences between groups. Kruskal–Wallis Test: This non-parametric test was used to assess whether there were differences in genetic distances across different groups of species based on marker types. All data were processed and visualized using RStudio and SPSS for ease of interpretation.

Sequence analysis for primer design and synthesis

Sequence alignment and ClustalW analysis were performed using BioEdit software (Version 7.01). Following multiple sequence alignments across different species, haplotypes and polymorphic loci were identified and counted. Primer design was carried out using Primer 5.0. Primer for ITS2 (Chiou et al., 2007): aqITSF: 5′-TGAACGCAAGTTGCGCCCCAAGCCT-3′ and aqITSR: 5′-TGGGGTCGCGATGCGCACTATGATT-3′. Primer for trnL-trnF (Taberlet et al., 1991): trnS1:5′-GTTACTTATCTTTCCCATTC-3′ and trnA1:5′-TCCCGACCATTACCAA-3′. Primer for matK (Yu, Xue & Zhou, 2011): CXS1: 5′-ATCCGCTGTGATAATGAG-3′ and CXS2: 5′-GCTTTCCGAACTTGGTTC-3′. For DNA barcoding analysis, each sequence was aligned with the sequences retrieved from NCBI. The homologous regions were identified and extracted for the design of universal primers. To ensure the stability and consistency of amplification, primers with lengths ranging from 18 to 25 nucleotides were designed in this study. The designed primers were synthesized by BGI Hong Kong, and the primers, with a molecular weight corresponding to 2 OD, were provided in 1.5 mL EP tubes.

PCR reaction and DNA sequencing

PCR Reaction: A total volume of 25 µL PCR reaction mixture was prepared, consisting of 12.5 µL of 2 × Premix PCR PrimeSTAR® HS DNA polymerase (Takara Biotech Co., Beijing, China), 10 ng of template DNA, and 0.2 µM of each primer. The matK gene sequences were amplified using a forward and reverse primer designed based on the methods described previously. PCR amplification was performed using a Veriti 96-Well Fast Thermal Cycler (Applied Biosystems, Foster City, CA, USA) under the following thermal cycling conditions: an initial denaturation at 94 °C for 2–5 min, followed by 30 cycles consisting of denaturation at 98 °C for 10 s, annealing at 54 °C (temperature optimized according to the primer sequences) for 10 s, extension at 72 °C for 30 s, and a final extension step at 72 °C for 7 min. DNA Quantification and Gel Electrophoresis: The concentration of the DNA was determined using a NanoDrop spectrophotometer to measure the absorbance ratio at 260 nm/280 nm. The DNA template and PCR products were then analyzed by electrophoresis on a 1% agarose gel at 100 V for 22 min. After electrophoresis, the gel was visualized and photographed using a gel imaging system. Clear and distinct PCR bands were excised and the resulting PCR products were sent to BGI (Shenzhen, China) for bidirectional DNA sequencing.

Haplotype analysis and phylogenetic reconstruction

Sequences of ITS (ITS2), rbcL, trnL-trnF, and matK were obtained from Aquilaria and four other genera (Gyrinops, Gonystylus, Wikstroemia, and Phaleria) within the Thymelaeaceae family, retrieved from NCBI. The gene haplotypes were identified and screened using DnaSP (Version 4.20) and Phase (Version 2.1). Haplotype networks (Bandelt, Forster & Röhl, 1999) were generated using the Median-Joining method in Network (Arlequin, Version 3.5.2.2), with network files manually input in net format for visualization using the “Draw Network” function. To construct the phylogenetic tree, interspecific K2P distances were calculated using MEGA 7.0 software. Both maximum likelihood and Neighbor-Joining methods were applied (with bootstrap = 1,000), and the resulting phylogenetic tree was used to classify sequences into distinct groups. For time tree analysis, BEAST V1.10.4 was used, with sequence data formatted in NEX or FASTA formats. The taxa were grouped according to the ML or NJ phylogenetic tree, and the substitution model, base frequencies, clock type, and tree process were specified in BEAUti. MCMC parameters were set as follows: chain length = 10,000,000; echo = 10,000; log = 200. The analysis was executed in BEAST, and the generated XML file was processed to produce the tree file. TreeAnnotator was used to calculate the burn-in value (1% of the chain length/sampling ratio) and produce the maximum clade credibility (MCC) tree, which was visualized and edited using FigTree (Version 1.4.4). Research process of species identification of Aquilaria using matK sequence for DNA barcoding was shown in Fig. S2.

Divergence time analysis of Aquilaria species

To analyze the divergence time between different species of Aquilaria, the software BEAST v1.10.4 was used, requiring the configuration of the Java Runtime Environment (JRE) and Java Development Kit (JDK). Species tree construction was carried out using BEAST, and sequence alignment was performed in MEGA 7.0, where a.nex format file was generated for this purpose. The BEAUti tool was employed to prepare the sequence alignment file and generate the corresponding XML file. The taxa function incorporated various pedigree relationships, and the relaxed clock model was initially selected for time clock estimation. The coefficient of variation was analyzed in Tracer, and if the value was below 0.5, the strict clock model was adopted for further analyses. The Yule process was chosen for the tree prior. A total of four independent runs were conducted, each with 5 × 107 generations, sampling every 1,000 generations. A burn-in of 10% was applied, and the results of the four runs were combined using LogCombiner. Tracer was also used to evaluate the effective sample size (ESS) and compare model performance. The evolutionary tree with the highest confidence was generated using TreeAnnotator. For additional divergence time estimation, a RelTime analysis was performed on the preconstructed maximum likelihood (ML) tree using MEGA X (Kumar et al., 2018). Divergence times at all branching points were estimated under the general-time-reversible (GTR) + F + G4 model, determined using ModelFinder (Kalyaanamoorthy et al., 2017) according to the Akaike Information Criterion. Fossil-derived calibration times were obtained from TimeTree (Hedges, Dudley & Kumar, 2006) and applied to Thymelaeaceae and its sister families. Internal calibrations for Thymelaeaceae were based on published fossil records, with minimum and maximum constraints placed according to the estimated credibility intervals (CIs) provided for two selected species.

Results

Distribution survey of different Aquilaria species

Aquilaria species have abundant sources, ranging from the northeast of the South Asian subcontinent to the Indonesian archipelago and Papua New Guinea archipelago, with at least 20 species of Aquilaria genus plants distributed, all of which can produce agarwood (Kiet, Keßler & Eurlings, 2005; Wang et al., 2021). Reports suggested that apart from the Aquilaria genus, there are four other genera within the Thymelaeaceae family, including Gyrinops, Gonystylus, Wikstroemia, and Phaleria, that are also recognized as fragrant resin producers in the agarwood market (Galicia-Herbada, 2006; Zhang Yong-Ceng et al., 2015). So, it’s need for more accurately understand species distribution information and traditional classification characteristics in the genus of Aquilaria. This study collected and summarized a comprehensive set of information on the distribution of Aquilaria species from scientific papers and business exchange websites (Farah et al., 2018; Lee & Mohamed, 2016; Ng, Chang & Kadir, 1997) (The Plant List, http://www.theplantlist.org/), as presented in Table 1 and Fig. S3. The analysis indicates a high concentration of Aquilaria species in Southeast Asia, with the Philippines hosting nine species, Indonesia eight, and Malaysia five. China records four species, including A. yunnanensis, A. sinensis, A. agallocha and A. grandiflora. Species diversity increases moving from mainland China into Southeast Asia, peaking in the Philippines and Indonesia. Vietnam has three species, while Cambodia and Laos each have two. Myanmar reports only A. malaccensis, and Thailand has three species. Indonesia, with the most species, includes A. beccariana, A. hirta, A. malaccensis, A. microcarpa, A. cumingiana, A. filaria, A. secundana, and A. moszkowskii. Brunei reports three species, and the Philippines features nine, including A. malaccensis, A. cumingiana, and A. filaria. Singapore has three species, and Papua New Guinea records two. India and Bangladesh each report one species, A. khasiana and A. malaccensis, respectively. This distribution highlights Southeast Asia’s crucial role in the agarwood market, reflecting a complex ecological and evolutionary history with varying species diversity across regions.

Table 1 Aquilaria species growth in different countries based on reported information.

Number	Countries	Species of Aquilaria	
1	China	A. grandiflora, A. sinensis, A. yunnanensis, A. agallocha	
2	Vietnam	A. crassna, A. banaense, A. malacensis	
3	Cambodia	A. crassna, A. baillonii	
4	Laos	A. crassna, A. malaccensis	
5	Myanmar	A. malaccensis	
6	Thailand	A. crassna, A. malaccensis, A. subintegra	
7	Malaysia	A. beccariana, A. hirta, A. microcarpa, A. rostrata, A. malaccensis	
8	Indonesia	A. beccariana, A. hirta, A. malaccensis, A. microcarpa, A. cumingiana, A. filaria, A. secundana, A. moszkowskii,	
9	Brunei	A. beccariana, A.cumingiana, A.grandifolia	
10	Philippine	A. malaccensis, A. cumingiana, A, filaria, A. brachyantha, A. urdanetensis, A. citrinaecarpa, A. apiculata, A. parvifolia, A. decemcostata,	
11	Singapore	A. hirta, A. malaccensis, A. macrocarpa	
12	Papua New Guinea	A. filaria, A. tomentosa	
13	India	A. khasiana	
14	Bangladesh	A. malaccensis	
15	Bhutan	A. malaccensis	

A survey of candidate sequences for molecular evolution and DNA barcoding in Aquilaria

Delving into molecular evolution, the investigation seeks to delineate the evolutionary trajectory of the targeted Aquilaria species. Four DNA barcoding markers were utilized in this study: ITS (ITS2), matK, rbcL, and trnL-trnF, to facilitate accurate identification and assessment of genetic diversity among 12 distinct Aquilaria species, which were downloaded from the NCBI database (summarized in Table 2). Key findings from the analysis include the predominance of A. sinensis, which yielded the highest number of sequences across all markers: 54 for ITS2, 23 for matK, 20 for rbcL, and 24 for trnL-trnF. Following A. sinensis, other notable species included A. crassna, A. malaccensis, and A. hirta, which are significant due to their cultivation in various regions from the northeast of the South Asian subcontinent to the Indonesian, as illustrated in Fig. S3. Intraspecies genetic variability was particularly pronounced in A. sinensis, with multiple SNPs identified across the sequences: seven SNPs in ITS2, six in rbcL, and eight in trnL-trnF. The matK sequence, however, displayed only two SNPs, indicating less variation in this region. The ITS2 sequence stood out as the primary focus for plant taxonomists, underlining its relevance in genetic studies and species identification. This study also acknowledged the challenge posed by the limited number of available sequences for certain species, exemplified by A. filaria, which had only a single sequence. This limitation emphasizes the need for further research and data collection to bolster the genetic understanding and conservation efforts associated with these valuable Aquilaria species. Based on ITS (ITS2), matK, rbcL, and trnL-trnF sequences, this study employs the Kimura 2-parameter (K2P) distance algorithm combined with Neighbor-Joining (NJ) tree construction to evaluate the capacity of these sequences for identifying species within the genus Aquilaria (as shown in Figs. 1–2). Unknown positions are treated as missing data. Bootstrap analysis is performed on each branch’s confidence levels, with a repetition count of 1,000 iterations. The results indicate that NJ trees constructed individually using ITS2 and rbcL sequences (e.g., Fig. 1A and Fig. 2A, Tables S6–S9) fail to effectively distinguish different Aquilaria species. For example, in Fig. 1A, the ITS2 sequence of A. malaccensis is distributed across the first three major clades and does not cluster properly together. Similarly, in Fig. 2A, the rbcL sequence of A. sinensis appears in different branches and does not cluster effectively. However, in Fig. 1B, except for A. malaccensis and A. yunnanensis, the matK sequences of other Aquilaria species can be reasonably differentiated. In Fig. 2B, the trnL-trnF sequences of different Aquilaria species demonstrate strong discriminative capability, with bootstrap support ranges between 30% and 95%. Specifically, in Fig. 2B, A. microcarpa, A. malaccensis, and A. beccariana form a clade with bootstrap support of 33%–62%; A. cumingiana and A. crassna form another clade with a bootstrap support of 30%. Additionally, A. rugosa, A. agallochum, A. sinensis, and A. yunnanensis form a clade with bootstrap supports ranging from 60% to 95%, indicating close phylogenetic relationships among these species with genetic differences. Notably, the trnL-trnF sequence of A. hirta forms its own branch with a bootstrap support of 52%, suggesting unique genetic characteristics for this species in this gene region.

Table 2 The information of candidate molecular markers sequences for Aquilaria including sequence number, haplotypes and SNP%.

	ITS Sequence	matK Sequence	rbcL Sequence	trnL-trnF Sequence	
Species	Sequence number	Haplotypes	SNP %	Sequence number	Haplotypes	SNP %	Sequence number	Haplotypes	SNP %	Sequence number	Haplotypes	SNP %	
A. sinensis	54	7	3.4	23	2	0.35	20	6	0.68	24	8	4.7	
A. crassna	41	3	0.5	31	1	0	32	1	0	36	4	0.6	
A. malaccensis	37	10	2.6	8	2	0.59	8	6	1.5	11	5	1	
A. microcarpa	14	4	1.25	4	1	0	—	—	—	5	2	0.4	
A. hirta	11	2	0.25	5	2	0	5	3	1	1	1	0	
A. subintegra	6	1	0	—	—	—	—	—	—	—	—	—	
A. beccariance	6	4	4	3	2	0.23	3	2	0.16	5	4	0.73	
A. yunnanensis	4	1	0	9	1	0	10	3	0.49	7	4	1.6	
A. agallocha	2	2	0.53	1	1	0	1	1	0	1	1	0	
A. cumingiana	2	2	0.5	1	1	0	1	1	0	1	1	0	
A. rugosa	2	2	0.5	1	1	0	1	1	0	1	1	0	
A. filaria	—	—	—	—	—	—	—	—	—	1	1	0	
Notes.

A., Aquilaria.

Figure 1 Comparison of different molecular markers (ITS and matK) used to study the genetic distances between different species of Aquilaria.

(A) ITS. (B) matK. The comparison includes measures of genetic distances, including interspecies distances and intraspecific distances. All ITS sequences used in this study refer to ITS2 sequence in ITS sequence. Bootstrap is repeated for 1,000 times, and the number in the figure is the self expanding support rate.

Figure 2 Comparison of different molecular markers (rbcL and trnL-trnF) used to study the genetic distances between different species of Aquilaria.

(A) rbcL. (B) trnL-trnF. The comparison includes measures of genetic distances, including interspecies distances and intraspecific distances. Bootstrap is repeated for 1000 times, and the number in the figure is the self expanding support rate.

This study evaluated the genetic variation within and between Aquilaria species using four sequence markers. As detailed in Tables S10 and S11, genetic distances were calculated using the K2P model and sequence alignment performed with MEGA 7.0. Intraspecific genetic distances based on the ITS2 sequence ranged from 0 to 0.0272 (mean = 0.0088), while interspecific distances ranged from 0.0265 to 0.0508 (mean = 0.0403), with interspecific distances approximately five times greater. A. crassna showed the highest intraspecific variation, while A. microcarpa, A. subintegra, and A. yunnanensis showed the lowest (all 0). For the matK sequence, intraspecific distances ranged from 0 to 0.0092 (mean = 0.0012) and interspecific distances ranged from 0.0046 to 0.0146 (mean = 0.0083), with interspecific variation seven times higher. A. sinensis exhibited the largest intraspecific distance, while several species had 0 distances. For rbcL, intraspecific distances ranged from 0 to 0.0049 (mean = 0.0015) and interspecific distances ranged from 0.0017 to 0.0132 (mean = 0.0016), with minimal interspecific variation. A. hirta showed the highest intraspecific distance. Lastly, for the trnL-trnF sequence, intraspecific distances ranged from 0 to 0.0174 (mean = 0.0069) and interspecific distances from 0.0130 to 0.0658 (mean = 0.0424). In this case, interspecific genetic distances were approximately six times greater than intra-specific distances, with the highest intra-specific genetic distance observed in A. beccariana. In contrast, A. agallochum, A. cumingiana, A. subintegra, and A. rugosa all exhibited intra-specific distances of 0. As illustrated in Fig. 3A, the intraspecific genetic distances based on the ITS sequence were significantly greater than those observed for the matK and rbcL sequences. Figure 3B shows that interspecific genetic distances based on the ITS and trnL-trnF sequences were notably higher than those for the matK and rbcL sequences. According to the “10 × rule” proposed by Hebert et al. (2003) for DNA barcoding, interspecific genetic distances must exceed 10 times the average intra-specific genetic distance to be considered sufficient for species differentiation. Based on the ITS, matK, and trnL-trnF sequences, the interspecific genetic distances (5–7 times the average intra-specific distances) meet the criteria for species differentiation. However, the rbcL sequence, with interspecific distances similar to intra-specific distances, does not fulfill the threshold required for species differentiation.

Figure 3 The comparison of different molecular markers sequences of Aquilaria using interspecies distances, and intraspecific distances.

(A) Box plots of interspecies distances alignment analysis for ITS, matK, rbcL, and trnL-trnF sequences. (B) Comparison bar chart of inter species distances for ITS, matK, rbcL, and trnL trnF. T-Test, *P < 0.05, N > 10.

The molecular identification technologies developed for Aquilaria and agarwood products

The development of advanced molecular identification techniques for Aquilaria and agarwood products has enabled the precise characterization of genetic diversity across these species. Specific primers were designed to target key regions, including ITS (ITS2), matK, and trnL-trnF sequences, ensuring reliable amplification with minimal contamination. As illustrated in Fig. 4A, eight highly polymorphic loci (+67, +68, +92, +161, +173, +199, +233 and +253) were successfully selected within the ITS2 sequence for effective species discrimination (Fig. 4A). The trnL-trnF sequence analysis revealed two notable polymerase sliding regions (+32∼44 and +323∼343) characterized by poly(-T-)n structures. These structural features significantly impacted sequencing efficiency, as evidenced by Fig. 4B. The optimized primer design confirmed the presence of low haplotype diversity across agarwood samples, with only three polymorphic loci (+173, +183 and +200) identified in all collected samples (Fig. 4C). Furthermore, an extensive polymorphism profile was observed within the matK sequence, encompassing at least eight stable polymorphic loci. Through detailed analysis of key polymorphic loci identified across the matK sequence, three critical loci (+249, +435 and +684) were pinpointed as highly discriminative markers for identifying medicinal (A. sinensis) in traditional Chinese medicine. The specific polymorphisms at these loci provide clear diagnostic criteria: At locus +249, consistent cytosine bases across all A. yunnanensis and A. sinensis samples indicate a Chinese mainland origin. At locus +435, the presence of guanine ensures identification as A. sinensis. At locus +684, the unique thymine base pattern further solidifies recognition of A. sinensis.

Figure 4 Design of specific primers and amplified fragments for Aquilaria identification.

(A) Specific primers design and the amplified fragment of ITS sequence for the effective region of polymorphic site concentration. (B) Specific primers design and the amplified fragment of trnL-trnF sequence for effectively avoiding sequencing abnormalities. (C) Specific primers design and the useful loci on matK sequence for Aquilaria identification.

Specific amplification targeting these critical polymorphic regions within the Aquilaria genus offers significant utility for species discrimination using matK-derived universal primers. As demonstrated in Fig. 4C and Table S12, this approach achieved highly sensitive identification with minimal cross-reactivity. The DNA barcode system provides a robust molecular identifier that enables rapid distinction between Chinese agarwood (A. sinensis) and imported samples based on distinct sequence characteristics. Analysis of 34 commercial sources agarwood revealed a strong correlation between the polymorphic loci (+684, +249 and +435) and Aquilaria species distribution. Among these samples, 64% (22/34) were identified as Chinese agarwood, predominantly sourced from markets within China or Cambodia (Table S12). PCR results confirmed the presence of distinct molecular band patterns, with imported agarwood exhibiting a characteristic 600 bp band and Chinese agarwood showing a more compact ∼300 bp pattern (Fig. S4). The DNA barcode system derived from matK sequence polymorphisms offers not only high accuracy but also significant practical advantages in terms of speed and ease of use. Further analysis employing genetic distance metrics calculated across seven key loci (+58, +124, +232, +261, +338, +373 and +559) within matK sequences revealed three distinct phylogenetic clusters when applying both NJ tree and ML tree methodologies. A notable finding is the division of Chinese agarwood samples into two categories, prompting a reevaluation of distribution data and recognizing the presence of A. agalloca in China (Fig. S5). To enhance the classification and assessment of sample origins, polymorphism information from an additional set of three loci on the trnL-trnF sequence was incorporated. Agarwood samples from China, Vietnam, Indonesia, Cambodia and other places can be well clustered together in their respective regions of origin, although a small portion of products from Vietnam, Cambodia, and Indonesia have been identified as A. senensis (Fig. S6). Continued analysis confirmed the earlier findings regarding the high prevalence of Chinese mainland Aquilaria in the sample population, with two additional phylogenetic groups identified among the remaining samples: one corresponding to Malaysian Aquilaria and another to Indonesian Aquilaria.

Analysis of the evolutionary status and genetic background of Aquilaria

Based on the maximum likelihood (ML) tree constructed using the matK sequence across 13 haplotypes representing 10 species within the Aquilaria genus, the divergence between Thymelaeaceae and Wikstroemia was estimated to have occurred approximately 60.7 million years ago (Ma) (Fig. 5). Within Thymelaeaceae, the divergence between Octolepidoidae and Thymelaeoideae was estimated at around 42.57 Ma. The earliest divergence in Aquilarieae was identified as the common ancestor of A. hirta (6.78 Ma), with separation from the other ingroup species occurring approximately 3.45 Ma. More recent divergence events were observed between A. malaccensis and A. beccariana (0.86 Ma), followed by the divergence between A. sinensis and A. agallochum (0.90 Ma) (Fig. 5). The analysis also revealed that three haplotypes of A. sinensis (H01, H02, and H03) exhibited distinct placements on different branches of the phylogenetic tree, with A. yunnanensis forming an independent branch adjacent to A. sinensis. Notably, A. agallochum and A. rugosa were closely related to A. sinensis, belonging to the same lineage. Additionally, as shown in Fig. 5, four distinct lineages were observed, referred to as the Chinese pedigree (C1 and C2), the Indonesian pedigree (B), and the Indochina pedigree (A), corresponding to A. sinensis, A. malaccensis, and A. cumingiana, respectively. Regarding the evolutionary trajectory, A. malaccensis (0.86 Ma) was identified as the most recent species, closely related to A. sinensis (0.90 Ma). This finding corroborates a previous study by Farah et al. (2018) and suggests that A. sinensis and A. malaccensis may have a significant advantage in terms of genome size compared to other species in the genus.

Figure 5 Divergence tree constructed by independent molecular characteristics on matK sequence in the genus Aquilaria.

Furthermore, a Median-Joining network map was generated to visualize the haplotypes of the matK sequences (Fig. 6 and Fig. S7). By combining the haplotype linkages in Fig. 6 with the species divergence time estimates in Fig. 5, it is evident that A. hirta (6.78 Ma), located in Malaysia, represents the earliest Aquilaria species. When it evolved into A. crassna, there were two base substitutions in the matK haplotype sequences. The haplotypes A. rugosa_ 01, A. cumingiana_01, and A. beccariana_01 all evolved from a common haplotype, A. crassna_01. Notably, the transition from A. crassna_01 to A. rugosa_01 involved a relatively higher frequency of base substitutions, with a total of four substitutions. In contrast, the transitions between other haplotypes typically involved only one base substitution. Through the evolution of haplotype A. rugosa_01, it eventually gave rise to A. sinensis_02 (with A. sinensis being the youngest species at 0.86 Ma, located in China). Similarly, the haplotype A. beccariana_ 01 evolved into A. malaccensis_01 (with A. malaccensis being the youngest species at 0.90 Ma, found in Southeast Asia).

Figure 6 The Median-Joining map constructed by using matlK haplotypes for Aquilaria genius analysis.

Each colored circle represents a haplotype, the size of the circle represents the number of samples contained in the haplotype, the line between the two circles indicates that there is a certain mutation connection between the two haplotypes, and the short line on the line represents the number of base substitutions required to change from one haplotype to another.

Discussion

Distribution patterns and ecological significance of Aquilaria species

Aquilaria, a plant of significant economic and medicinal value, has been a focal point for researchers worldwide aiming to safeguard its genetic resources and establish a scientific foundation for sustainable development and utilization (Kang, 2021; Rasool & Mohamed, 2016; Wang et al., 2021). This global effort has resulted in extensive investigations into the diversity of Aquilaria germplasm resources and the population genetic structure (Banu et al., 2015; Salgotra & Chauhan, 2023; Singh et al., 2015). This study presents a comprehensive survey of the distribution of various Aquilaria species, shedding light on the species’ geographical patterns and their significance in the agarwood market. The results indicate that Southeast Asia is the epicenter of Aquilaria species diversity, with countries like the Philippines, Indonesia, Malaysia, and China hosting a variety of species, as detailed in Table 1. Notably, the Philippines leads with nine species, followed by Indonesia with eight, reflecting the critical role of these regions in the agarwood industry. China, with its four recorded species, holds unique significance, particularly due to the presence of species such as A. agallocha and A. grandiflora, which remain unfamiliar to many Chinese medical practitioners. This highlights a gap in local knowledge and suggests potential avenues for the further study and utilization of these species in traditional and modern medicine. A key finding of this study is the observed increase in species diversity as one moves from mainland China toward Southeast Asia, with the greatest species richness found in the Philippines and Indonesia. This pattern reflects not only ecological factors but also the complex evolutionary history of the genus. Morphological research emerges as a pivotal tool for swiftly assessing the genetic landscape through morphological traits, offering a precise analysis of variation. This method stands out as the most concise and direct approach for detecting genetic diversity, playing a crucial role in both interspecific and intraspecific classification (Pech-Hoil et al., 2023; Shah et al., 2023). The research presents the conservative distribution information of the five main cultivated Aquilaria species based on the collected specimens (Fig. S3). Among them, A. sinensis stands out with its significant quantitative advantage in mainland China. These findings emphasize the importance of understanding the taxonomy and diversity within Aquilaria species.

Insights into the genetic diversity and species differentiation of Aquilaria

With the rapid development of molecular biology and genetics research, the research on the genetic diversity of Aquilaria has gone from the morphological level to the molecular level. In the last decade, the application of DNA barcode technology has emerged as a robust method for identifying Chinese herbal medicines. Numerous laboratories have demonstrated its effectiveness in rapidly and reliably identifying these medicines, transcending their diverse appearances (Antil et al., 2023; Gong et al., 2018; Kress, 2017; Lin et al., 2023; Mahima et al., 2022). The operational procedure typically involves DNA extraction, sequence amplification using universal primers, sequencing, and subsequent database comparison. It is crucial to highlight that the accuracy of identification results is intricately tied to the quality of the sequence database (Burian et al., 2021; Gupta, 2019). However, amidst the success, several scientific oversights have surfaced, with a pivotal concern being the efficient and expeditious acquisition of a comprehensive genetic background of a species. This study focuses on investigating the evolutionary trajectory of Aquilaria species through molecular evolution analysis and DNA barcoding. Three specific segments derived from the chloroplast genome, namely, matK, rbcL, and trnL-trnF, and the ribosomal DNA internal transcribed spacer (rDNA-ITS) sequence ITS2 were selected as recommended molecular sequences for Aquilaria classification. The results reveal that A. sinensis, followed by A. crassna, A. malaccensis, A. hirta, and A. subintegra, are the most researched and economically significant species, cultivated across various regions. Among the sequences analyzed, ITS2 emerged as the most widely used marker for species identification, highlighting its importance for plant taxonomists. However, it was noted that the availability of sequences for certain species remains limited, with some species represented by only a single sequence, which may affect the reliability of the genetic data. The study also found notable differences in genetic variation across the species, particularly with regard to single nucleotide polymorphisms (SNPs) (Table 2). While A. sinensis displayed substantial intraspecific variation, the rbcL and ITS2 markers failed to provide adequate resolution for species differentiation. The findings suggest that these markers, while useful, may not be sufficient on their own for accurate species identification. In contrast, the matK sequence provided better differentiation among Aquilaria species, particularly in comparison to rbcL and ITS2. This is in line with the observed genetic distances, where matK exhibited a higher intraspecific variation compared to rbcL (Fig. 3), suggesting that it is a more effective marker for distinguishing species within the genus. The comparison of interspecific and intraspecific genetic distances, particularly through the use of the “10 × rule” for DNA barcoding (Hebert et al., 2003), reveals that the ITS, matK, and trnL-trnF sequences fulfill the necessary criteria for species differentiation, with interspecific distances being significantly higher than intra-specific distances. In contrast, rbcL did not meet this threshold, demonstrating its limitations in species identification. This study introduces a novel approach to evaluating the efficacy of molecular markers by comparing their ability to distinguish between species, providing valuable insights into the performance of DNA barcoding markers in Aquilaria research. The findings underscore the need for further refinement of DNA barcoding protocols and the expansion of sequence databases to include more comprehensive data for underrepresented species, such as A. filaria. Additionally, the study highlights the potential of the matK marker as a more reliable tool for species differentiation in the Aquilaria genus, particularly in cases where other markers, such as ITS2 and rbcL, may not provide sufficient resolution. Further research into the genetic diversity of Aquilaria species, particularly with regard to highly polymorphic regions, is crucial for advancing molecular identification techniques and enhancing conservation efforts for these economically valuable species.

Multifaceted approach to molecular identification of Aquilaria and agarwood products

The exploration of molecular taxonomy for Aquilaria and agarwood products reveals the necessity of a comprehensive approach that extends beyond the use of a single sequence, such as matK. Various studies have underscored the importance of adopting a multifaceted strategy for accurate species identification (Austen et al., 2016; Baerwald et al., 2020; Pereira et al., 2021; Sun, Futahashi & Yamanaka, 2021; Wäldchen et al., 2018). The findings from this study reinforce the importance of integrating various molecular markers, such as the ITS2 and trnL-trnF sequences, to improve species identification accuracy. Our results indicate that the matK sequence, with its eight stable polymorphic loci, offers significant utility for distinguishing between different Aquilaria species, particularly in the identification of medicinal agarwood varieties used in traditional Chinese medicine. The specific loci at positions +249, +435, and +684 serve as highly discriminative markers, with locus +684 featuring a unique thymine base pattern that reliably identifies A. sinensis (Fig. 4). Similarly, the presence of guanine at locus +435 is crucial for recognizing A. sinensis, while cytosine at locus +249 is indicative of Chinese mainland-origin Aquilaria species. The analysis of the trnL-trnF sequence, although showing limited polymorphic variation, provides additional context for species differentiation. However, it is clear that further exploration of the trnL-trnF sequence is necessary to fully harness its potential for molecular identification. This gap highlights the importance of continuing to develop and refine genetic markers to improve the breadth and sensitivity of species detection in Aquilaria. Further investigation of 34 commercial agarwood sources revealed a strong correlation between the specific loci (+684, +249, +435) and the geographical origins of Aquilaria species. This was confirmed through PCR-based assays, which produced distinct molecular band patterns for Chinese versus imported agarwood, underscoring the value of targeted PCR sequencing in confirming the authenticity and origin of agarwood products. The presence of a 600 bp band in imported samples and a 300 bp band (Fig. S4) in Chinese agarwood samples provides a clear, reliable diagnostic tool for identifying legally recognized medicinal materials. Additionally, the clustering analysis of matK sequences across 34 agarwood samples revealed three distinct phylogenetic groups, consistent with known distribution patterns of Aquilaria species (Fig. S5). This analysis not only corroborates the earlier findings but also identifies new insights into the species’ distribution, including the recognition of A. agalloca within the Chinese mainland population. These findings further demonstrate the complexity and diversity of agarwood products, highlighting the need for more refined molecular tools to address variations in both origin and species classification.

Evolutionary insights and implications for Aquilaria apecies classification

As navigating the complexities of Aquilaria genetics and taxonomy, it is essential to contextualize our findings within the broader body of research in this field. Several studies have underscored the importance of integrating molecular data with ecological and geographical information to elucidate patterns of species divergence and evolution (Aguirre-Liguori, Ramírez-Barahona & Gaut, 2021; Voet et al., 2022; Zhou, Xiang & Wen, 2020). Additionally, recent advancements in next-generation sequencing technologies have opened new avenues for exploring genetic diversity and population structure within Aquilaria species (Hishamuddin et al., 2020; Hu et al., 2021; Lee et al., 2022; Li et al., 2019; Satam et al., 2023). The molecular clock analysis of 13 haplotypes from 10 species has revealed crucial divergence times and evolutionary events that shape the genetic structure of this genus. The findings reinforce previous studies, such as Farah et al. (2018), which identified A. hirta as the oldest member of Aquilaria, with a divergence approximately 6.78 Ma. This supports the hypothesis that A. hirta represents an early evolutionary branch. The focus on matK-based haplotype clustering of this study (e.g., Chinese vs. Indochinese lineages) could be linked to prior findings on biogeographic patterns. For example, Lee et al. (2022) identified rapid divergence events in Aquilarieae during the Pleistocene (∼1.24 Ma), which aligns with this study’s observation of recent speciation in A. sinensis and A. malaccensis (∼0.9–1.0 Ma) (Fig. 5). Additionally, the molecular clock analysis highlighted the formation of distinct lineages within Aquilaria, including the Chinese (C1 and C2), Indonesian (B), and Indochina (A) pedigrees, corresponding to A. sinensis, A. malaccensis, and A. cumingiana, respectively (Fig. 5). The close relationship between A. sinensis, A. agallochum, and A. rugosa within the same lineage aligns with the phylogenetic tree generated in our study, emphasizing their evolutionary proximity. One of the most compelling aspects of this research is the detailed divergence timeline within the genus, including the critical divergence event around 42.57 Ma within the Thymelaeaceae family and the more recent events between A. malaccensis and A. beccariana (0.86 Ma). The results of our Median-Joining network map (Fig. 6) and divergence time estimates offer a comprehensive view of the haplotype evolution and provide a clearer understanding of the genetic relationships within Aquilaria. Geographically, the analysis contradicts earlier hypotheses about the origin of Aquilaria. Contrary to previous suggestions that India was the origin center for this genus, this study strongly suggests that the Indonesian Islands represent the primary diversification hub for Aquilaria species. A. malaccensis and A. sinensis, as the dominant species in subtropical and tropical rainforests, are central to this geographic origin (Fig. 7). The results also clarify the distribution of A. rugosa, which is confined to China and the Indochina peninsula, further solidifying the notion of multiple, independent evolutionary lineages within the genus. This regional differentiation highlights the complexity of Aquilaria’s evolutionary trajectory, with distinct genetic lineages emerging in geographically isolated regions. The implications of these findings extend beyond taxonomy. By understanding the evolutionary history of Aquilaria, we gain insights into its ecological roles and adaptive strategies in different environments. This is particularly relevant for conservation efforts, as habitat fragmentation and climate change may affect the distribution and survival of these species. Moreover, the identification of distinct genetic lineages could aid in breeding programs and the sustainable harvesting of agarwood, ensuring genetic diversity is preserved. Furthermore, this study challenges traditional views on species classification within the genus. Specifically, we present strong evidence supporting the reclassification of G. walla into the Aquilaria genus. The morphological similarities between G. walla and A. malaccensis, along with its shared genetic lineage, suggest that G. walla should not be considered an early divergent species, as previously assumed (López-Sampson & Page, 2018). This reclassification has important implications for both taxonomic accuracy and the broader understanding of agarwood-producing plants.

Figure 7 Aquilaria species distribution and pedigree characteristics under the assumption of origin centre in Malacca.

Conclusions

This study emphasizes the pivotal role of molecular markers, particularly within the matK sequence, for accurately identifying and classifying Aquilaria species and agarwood products. The use of specific loci, such as guanine at position +435 and cytosine at +249 of matK sequence, enhances the resolution of species identification, especially for medicinal varieties. Additionally, the study explored the geographic origin of agarwood through molecular band patterns observed in 34 commercial samples, which revealed a clear distinction between imported samples (600 bp) and Chinese-origin agarwood (300 bp). From an evolutionary standpoint, molecular clock analysis of 13 haplotypes across 10 species highlighted A. hirta as the most ancient species, with an estimated divergence dating back approximately 6.78 million years. The genetic split between A. sinensis and A. malaccensis occurred around 0.86–0.90 million years ago, marking significant evolutionary milestones. Furthermore, four distinct genetic lineages were identified, corresponding to A. sinensis (C1-2), A. malaccensis (B), and A. cumingiana (A), suggesting that these species have multiple independent origins. Lastly, the reclassification of G. walla is proposed based on its genetic affinity with A. malaccensis, challenging previous morphological classifications and suggesting its incorporation into the Aquilaria genus. This reclassification provides new insights into the genetic diversity and evolutionary history of agarwood species.

Supplemental Information

Supplemental Information 1 Agarwood samples (wood) from different countries using matK sequence for DNA barcoding

Supplemental Information 2 Research process of species identification of Aquilaria using matK sequence for DNA barcoding

Supplemental Information 3 Regional distribution of five main cultivated species of Aquilaria genus

The triangle denotes A. crassna, the circle represents A. subintermedia, the prism symbolizes A. malaccensis, the sun-shaped symbol indicates A. hirta, and the pentagram signifies A. sinensis.

Supplemental Information 4 The original image of amplified fragments for Aquilaria identification with full length membranes

Supplemental Information 5 Phylogenetic analysis and systematic cluster analysis for agarwood products and Aquilaria using matK sequence only

Supplemental Information 6 Phylogenetic analysis and systematic cluster analysis for agarwood products and Aquilaria using matK and trnL-trnF sequence

Supplemental Information 7 Evolutionary tree analysis on different haplotypes of the Aquilaria genus

Supplemental Information 8 Supplementary tables

Supplemental Information 9 ITS sequence of Aquilaria

Supplemental Information 10 matK sequence of Aquilaria

Supplemental Information 11 maTk sequences of Aquilaria for time clock

Supplemental Information 12 rbcL sequence of Aquilaria

Supplemental Information 13 trnL sequence of Aquilaria

We would like to express our gratitude to Professor Zhou Hua and his team from the Faculty of Chinese Medicines, Macau University of Science and Technology, for their valuable support and collaboration.

Additional Information and Declarations

Competing Interests

Author Contributions

Data Availability

The authors declare there are no competing interests.

Zhaoqi Xie conceived and designed the experiments, performed the experiments, analyzed the data, prepared figures and/or tables, authored or reviewed drafts of the article, and approved the final draft.

Siqing Fan conceived and designed the experiments, performed the experiments, analyzed the data, prepared figures and/or tables, authored or reviewed drafts of the article, and approved the final draft.

Junyu Xu performed the experiments, analyzed the data, authored or reviewed drafts of the article, and approved the final draft.

Haijing Xiao analyzed the data, prepared figures and/or tables, and approved the final draft.

Jiaxin Yang conceived and designed the experiments, prepared figures and/or tables, authored or reviewed drafts of the article, and approved the final draft.

Min Guo analyzed the data, authored or reviewed drafts of the article, and approved the final draft.

Chunsong Cheng conceived and designed the experiments, analyzed the data, authored or reviewed drafts of the article, and approved the final draft.

The following information was supplied regarding data availability:

The raw measurements are available in the Supplemental Files.

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
