# Peer review of "Identification and classification of Aquilaria (Thymelaeaceae): inferences from a phylogenetic study based on matK sequences"

_PeerJ, doi:10.7717/peerj.19752_

## Round 0.1 · original submission · Major Revisions

Thank you very much for your manuscript titled “Identification and classification of Aquilaria (Thymelaeaceae): inferences from a phylogenetic study based on matK sequences” that you sent to PeerJ.

As you will see below, comments from referee 1 suggest a minor revision while reviewers 2, 3, and 4 suggest a major revision before your paper can be published. Given this, I would like to see a major revision dealing with the comments. Their comments should provide a clear idea for you to review, hopefully improving the clarity and rigor of the presentation of your work. I will be happy to accept your article pending further revisions, detailed by the referees, which largely focus in improving in various ways the different parts of the manuscript, from the abstract to the discussion.

Please note that we consider these revisions to be important and your revised manuscript will likely need to be revised again.

Reviewer 1 ·

Basic reporting

i) Language and Clarity:
The manuscript employs professional and clear English throughout. However, there are occasional instances where the phrasing could benefit from more precision to enhance readability (e.g., abstract lines 12-20). It is recommended to simplify and refine complex sentences for better comprehension.

ii) Background and References:
The introduction provides a comprehensive background on Aquilaria, with sufficient references to relevant literature. The cited studies are appropriate and recent, though the authors could further emphasize the novelty of their work in relation to existing research.

iii) Structure and Figures:
The manuscript adheres to PeerJ's standards for structure. Figures and tables are relevant, clearly labeled, and adequately described. However, Figure S3 (Supplementary) could benefit from better resolution for easier interpretation.

iv) Raw Data:
The raw data and supplementary materials are shared and seem comprehensive. However, a clearer explanation of how the raw data align with the results presented would improve transparency.

v) In formal academic writing, it is customary to avoid personal pronouns such as "we," our ," or "you" to maintain an objective tone. For example, instead of "We analyzed the data," consider using "The data were analyzed." Similarly, "You can see the results in Figure 1" could be revised to "The results are presented in Figure 1." Adjusting these phrases throughout the manuscript will enhance the professional tone and align with academic standards.

Experimental design

i) Scope and Originality:
The study fits within the journal's aims, addressing a well-defined research gap concerning the classification of Aquilaria species using molecular markers. The question is meaningful, with clear implications for conservation and commercial applications.

ii) Methods:
-The Materials & Methods section is comprehensive, but the dense text could make it challenging for readers to follow the workflow. To enhance readability and improve understanding, please add a flowchart that visually outlines the methodology. The flowchart could summarize key steps such as sample collection, DNA extraction, marker selection, sequence alignment, and phylogenetic analysis. -The authors should clarify:
The justification for selecting specific molecular markers like matK over alternatives.
Why trnL-trnF exhibited lower polymorphic loci and its implications for reliability.

iii) Ethical Standards:
The collection and identification of plant material align with ethical and professional norms. Voucher specimens deposited in a public herbarium enhance credibility.

Validity of the findings

-The underlying data appear robust and statistically sound. The use of molecular clock analyses and phylogenetic trees is appropriate and supports the study's conclusions.

-The findings are replicable, as the methods are described in detail. While the integration of matK and trnL-trnF markers is a valuable contribution, the novelty could be highlighted more explicitly, particularly in comparison to prior studies.

-Conclusions:
The conclusions are well-aligned with the original research question and supported by the results. The suggestion to include G. walla within the Aquilaria genus is intriguing and could spark further research.

Additional comments

The manuscript makes a significant contribution to the taxonomy and conservation of Aquilaria.

The figures and supplementary materials are valuable but could benefit from minor enhancements in clarity and resolution.

Reviewer 2 ·

Basic reporting

Is Adequate

Experimental design

Adequate

Validity of the findings

Sentences from 317 to 328 need to be simplified.
Finding section is not too the point, major section needs to me moved to discussion specially line no 341 to 371.

The result segment needs to be crisp and affirmative not speculative, as it is now. Reframing to results segment needed

Table 1:
There is no such country named Bengal???

Reviewer 3 ·

Basic reporting

This work, entitled “Identification and classification of Aquilaria (Thymelaeaceae): inferences from a phylogenetic study based on matK sequences”, according to the authors, is aimed "to try to sort out the evolutionary relationship between wild and cultivated Aquilaria species by molecular sequence and infer their frequency and mode of gene exchange in Asian tropical rainforest (lines 87-89)"...and..."At least genotyping (or genetic typing) of products of unknown species should be achieved, thus for further developing chemical indicators for quality control (lines 90-92)". Reading the title of the manuscript, the introduction and the aims, I get lost with what the authors want to capture.
1.- In the introduction, from line 33 to line 81, the authors imply the importance of classifying Aquilaria species and the distribution of populations, and highlight the need to find a method to identify the species due to their overexploitation, because there is no clear traceability of agarwood products. However, nowhere in the introduction do they report on current work in systematics, evolution and phylogeny of Aquilaria species. That is why I am lost with the manuscript because there is a mixture of phylogeny, evolution, identification, genetic traceability of processed products and search for quality control techniques. Therefore, the objective and hypothesis are not coherent.
In fact, the phylogeny and evolution of Aquilaria species is largely established with DNA Barcoding (ITS, matK, rbcL and trnL, among others; Farah et al 2018; doi: 10.3389/fpls.2018.00712) and complete chloroplast genomes (doi.org/10.1038/s41598-020-70030-0; Lee et al 2020; and doi.org/10.1093/botlinnean/boac014; Lee et al 2022)
2.- The literature is not well referenced, the authors cite papers from Researchgate (see lines 547-549 and 611-612) and others are not well done (lines 568-570). There are several uncited statements, for example one on lines 48-51.
3.- The structure of the article is not clear.
4.- In the results item, the authors explain most of their results as a qualitative rather than quantitative análisis (226-241; 242-261; 264-328), and many times cite papers in this item (Lines 208, 212 and 225), implying that it is in the discussion item. The results should be presented with quantitative values to give support and consistency.

Experimental design

In the first part of Material and Method (Plant material acquisition and identification) it is not clear what type of material was used (leaf, wood, etc) and which plant material was deposited in the herbarium. According to the authors, the list in table S1 and the list in table S2 were used in the study, but one list has 300 samples and the other has 34 samples, I think this paragraph is poorly worded. On the other hand, the authors say that the material was purchased and then identified. How to know for sure that what was purchased comes from that area of origin? This is where the work is weakened. The authors place the burden of identifying more than nine species of Aquilaria on one person, Dr. Chen Chun Song. However, there was a lack of rigor here, since one of the shortcomings of the work was not to have evidenced a taxonomic analysis of the samples obtained (photos of at least one representative per species (voucher) in the supplementary material), since these were not even collected, but purchased. The work does not explain the taxonomic keys of the traits that differentiate Aquilaria species.
1.-To perform genetic traceability, at least one sure positive control of each Aquilaria species sample is needed, but this is not observed in the paper.
2.-In lines 119-120 you have to detail which mathematical calculations were implemented.
3.-Samples processing and DNA extraction I think it should be before Molecular Marker sequence acquisition and analysis.
4.- The specific primers designed by the authors do not appear in the material and method (lines 153-160), but are shown in Figure 3, which corresponds to the Results item.
5.- The authors do not cite any work on the DNA sequence (ITS, matK, etc) they used to design their specific primers.
6.- The divergence analysis I suggest eliminating it from this work, especially because it is already established in other publications (Farah et al 2018 doi: 10.3389/fpls.2018.00712; Lee et al 2022 doi.org/10.1093/botlinnean/boac014). In addition, the analysis (lines 186-198) is not well elaborated because they did not use calibration (the authors did not use a reference fossil in the analysis).
7.- The authors point out that the ITS is a segment of the Chloroplast genome (lines 406-408), this is false. Which ITS did the authors use, ITS1 or ITS2?

Validity of the findings

1.-I believe that the identification and replication of the 34 agarwood products is not clear. The technique is not well established, because the authors should perform an exhaustive identification of each species, which was not achieved. On the other hand, agarwood products do not only come in wood, but there are also products such as oils, among others.
2.-Barcoding DNA sequences of Aquilaria species had already been discovered by other authors.
3.- According to the above, the conclusion of this manuscript is not clear.

Additional comments

A deep redirection of the manuscript to genetic traceability of agarwood products is required, but not to be mixed with systematics.

Reviewer 4 ·

Basic reporting

The introduction would benefit from a clearer focus on the main research questions and the supporting information that addresses them. In its current form, the narrative is somewhat vague, making it unclear how the discussion aligns with or targets the core research objectives.

Experimental design

The methodology section requires further detail to enhance both readability and, more importantly, reproducibility. Specifically, references and detailed descriptions are missing for nearly all of the software utilized in the study. Providing comprehensive information about the software, including versions, settings, and relevant references, will enable readers to better understand and replicate the analyses. Without this, the study's practicality and the reliability of its results may be significantly compromised.

Validity of the findings

Many of the methods described in the Materials and Methods section lack proper citations. It is essential to reference the original papers or sources where these methods were previously applied to establish the validity and reliability of the results and analyses.

Additional comments

This study focuses on identification Aquilaria (Thymelaeaceae): using a phylogenetic framework. I find the study interesting and of valuable to the field. However, there are some issues and concerns that need to be addressed prior to publication.

Major comments:

1. The methodology section requires further detail to enhance both readability and, more importantly, reproducibility. Specifically, references and detailed descriptions are missing for nearly all of the software utilized in the study. Providing comprehensive information about the software, including versions, settings, and relevant references, will enable readers to better understand and replicate the analyses. Without this, the study's practicality and the reliability of its results may be significantly compromised.

2. The introduction would benefit from a clearer focus on the main research questions and the supporting information that addresses them. In its current form, the narrative is somewhat vague, making it unclear how the discussion aligns with or targets the core research objectives.

3. Many of the methods described in the Materials and Methods section lack proper citations. It is essential to reference the original papers or sources where these methods were previously applied to establish the validity and reliability of the results and analyses.

Abstract

Line 14-15: please give some examples for those 8 stable polymorphic loci.

Line 25-27: Please mention your take-home message for this research here with clear implications.

Introduction

Line 37-38: please add more information about why the quality is substandard, this sentence is vague.

Line 38-40: reference is needed.

Line 43-45: Having said that what is the over-arching goal of the study? Please mention it here.

Line 48-52: References are needed here.

Line 45-63: The purpose of this paragraph is not clear! Please start it with a clear explanation. Is it referring to genetic diversity and background? If so, a better opening sentence is needed.

Line 86: the sentence “the objectives of the conclusion” is vague, please rephrase it.


Material and methods:

Line 98-99: more information is needed here, a broad spectrum of agarwood types based on genetic variation or geographical origin?

Line 99-102: this information should not be here, please add it to the authorship section or put it in the acknowledgment section.

Line 108-111: please add more information on how the data were extracted from website. The sites mentioned are just the main website of those botanic garden. The procedure used should be clear and repeatable.

Line 113-115: how the coordinates were extracted using AMAP? Any packages? Website? Or codes in R. please be specific and clear. Reference is also needed here.
Lines 115-120: References are needed here.

Line 127: Please provide details about how BioEdit was used! Is there any version for this software? How can readers have access to it?


Results:

Line 203-205: it is the first time, the plant list is mentioned. Please add this to the M&M. How did you check? Direclty from website? Or R packages like Package WorldFlora were used?

Discussion:

Discussion is well-written but I recommend using subsections to better focus on each question/goals and present them in a well-organized fashion.


Figures:

The caption of each figure should be self-explanatory so readers can understand each figure without looking back at the text. Please add more information.

---

## Round 0.2 · accepted · Accept

After reviewing this revised version of your manuscript, I see that the comments suggested by the reviewers have been included, Therefore, I am satisfied with the current version and consider it ready for publication..

Reviewer 3 ·

Basic reporting

Dear authors and editor,
The authors have improved the manuscript and have adequately addressed the comments. At least from my perspective, in its current form, it is suitable for acceptance.

Experimental design

Dear authors and editor,
The authors have improved the manuscript and have adequately addressed the comments.

Validity of the findings

Dear authors and editor,
The authors have improved the manuscript and have adequately addressed the comments.

Reviewer 4 ·

Basic reporting

Clear English

Experimental design

Methods described with sufficient detail & information to replicate.

Validity of the findings

Conclusions are well stated, linked to original research question.